# Factors Associated with Vaccination Adequacy in People Living with HIV: A Cross-Sectional Study

**DOI:** 10.3390/vaccines12091003

**Published:** 2024-09-01

**Authors:** Larissa Gerin, Andrey Oeiras Pedroso, Marcela Antonini, Elucir Gir, Bruno Spire, Renata Karina Reis

**Affiliations:** 1Epidemiological Surveillance Office at Ribeirão Preto Municipal Health Department, Ribeirão Preto 14015100, São Paulo, Brazil; 2Ribeirão Preto College of Nursing, University of São Paulo, Ribeirão Preto 14040902, São Paulo, Brazil; apedroso@usp.br (A.O.P.); antonini.enf@gmail.com (M.A.); egir@eerp.usp.br (E.G.); rkreis@eerp.usp.br (R.K.R.); 3Inserm, IRD, SESSTIM, Sciences Economiques & Sociales de la Santé & Traitement de l’Information Médicale, ISSPAM, Aix Marseille University, 13385 Marseille, France; bruno.spire@inserm.fr

**Keywords:** vaccination coverage, vaccination, HIV

## Abstract

People living with HIV (PLHIV) are at greater risk of illness and death from vaccine-preventable diseases. This study aimed to identify the predictors associated with the recommended vaccination schedule for this group. This was a single-center cross-sectional study conducted in a large Brazilian municipality, evaluating the vaccination statuses of 645 PLHIV for nine immunizers. The primary outcome was the adequacy of the vaccination schedule. The vaccination status was assessed for the diphtheria/tetanus, hepatitis B, hepatitis A, measles/mumps/rubella, yellow fever, 13- and 23-valent pneumococcal, meningococcal C, and HPV vaccines. Those who had received all of the recommended vaccinations in accordance with the schedule established by the government at the time of the assessment, without any delays, were classified as having received an “adequate schedule”. The independent variables included sociodemographic, clinical–epidemiological, and social vulnerability factors, analyzed by multiple logistic regression with adjusted odds ratios (ORs) and 95% confidence intervals (95% CIs). Only 47 individuals (7.3%) had an adequate vaccination schedule for all vaccines. The vaccines with the highest adequacy rate were diphtheria and tetanus (533; 82.6%), and the one with the lowest rate was measles/mumps/rubella (MMR) (243; 37.7%). The main predictors of a complete vaccination schedule were the age group, place of clinical follow-up, and where they received their last doses of vaccines. Educational interventions for PLHIV and health professionals are needed to improve the vaccination coverage in this group.

## 1. Introduction

Vaccination constitutes the main public health intervention for the control of communicable diseases. It is a safe and effective strategy that has resulted in a notable decline in the mortality rates associated with infectious diseases. Furthermore, vaccines can prevent disabilities that may impair children’s growth and cognitive development. They also benefit adults and the elderly by preventing infection-related cancers and protecting people’s health, thus allowing them to live longer and healthier lives. In addition, vaccination is widely considered an important strategy for emerging infectious diseases, by containing or limiting disease outbreaks and preventing the spread of antimicrobial resistance [1].

While the advantages of vaccination for population health are clear, the decline in vaccination adherence and coverage observed globally in recent years represents a significant concern. Vaccine hesitancy, as identified by the World Health Organization (WHO), is among the top 10 global threats to public health [2,3,4]. 

The Brazilian government makes immunizers available to the population free of charge through the Unified Health System (UHS) in order to meet a national vaccination schedule that includes different indications for specific age groups. These groups include children, adults, pregnant women, puerperal women, the elderly, and other groups with special indications. In 2023, Brazil celebrated the 50th anniversary of its National Immunization Program (NIP), which is responsible for the National Immunization Policy. The latter aims to reduce the transmission of vaccine-preventable diseases and the occurrence of serious cases and deaths and to promote and protect the health of the Brazilian population by strengthening integrated surveillance actions [5]. 

The actions of the NIP have enabled the control of numerous diseases within the country, including smallpox and polio eradication. However, there has been a decline in vaccination coverage, which has resulted in an increased risk to the population’s safety [5].

Globally, immunization activities and efforts to achieve vaccination coverage are heavily focused on children, especially the youngest. Little is known about adult immunization coverage, and studies show low coverage in this group. This fact contributes to the lack of data to guide programs and governments in developing campaigns aimed at improving these indicators and facilitating access to immunizers, since there is a high burden of infectious diseases in this group [6,7,8]. 

People living with human immunodeficiency virus (PLHIV) may have a worse prognosis for vaccine-preventable diseases, and some diseases have a similar route of transmission to HIV [6,9]. 

On the other hand, modern treatments using antiretroviral therapy (ART) can ensure that the individual maintains an undetectable viral load undetectable, which reduces the possibility of immunosuppression, guaranteeing quality of life similar to that of people not affected by HIV, with the result that there is also greater exposure to infectious diseases [10,11,12].

It is recommended that the vaccination program be extended beyond those offered to the general population to enhance the security of this group [13,14]. In Brazil, in addition to the four vaccines made available to adults (diphtheria/tetanus, hepatitis B, yellow fever, measles/mumps/rubella), PLHIV are recommended to receive eight other vaccines (hepatitis A, HPV, meningococcal ACWY, 13- and 23-valent pneumococcal, *Haemophilus influenzae* type b, varicella, and influenza) through Special Immunobiological Reference Centers (SIRC) [13].

Despite the heightened vulnerability of PLHIV to infectious diseases and special indications for vaccination, the services responsible for the clinical follow-up of these individuals have failed to conduct an effective assessment of their vaccination status. Moreover, there is a lack of knowledge regarding the vaccination coverage in this group and the factors that can influence their immunization rates [6,10,11]. 

This is a very complex assessment given the number of immunizations currently on offer, with different schedules and indications that can vary according to the age group, susceptibility, and immune status of individuals. The potential contraindication of attenuated vaccines, which comprise live pathogens, in the context of severe immunosuppression, may contribute to the uncertainty of healthcare professionals in administering vaccines to this population [13,14].

Further, there is concern regarding the indication of vaccines for HIV PLHIV, given the limited availability of data on immunogenicity and efficacy in this population, which may contribute to the hesitancy in vaccination uptake. Despite the theoretical possibility of a reduced response, the available evidence suggests that the vaccination of PLHIV is beneficial in reducing hospitalizations and deaths from vaccine-preventable diseases [3,15,16,17]. 

Given this scenario, this study aimed to identify the predictors that can influence the vaccination schedules of PLHIV, taking into account nine vaccines indicated by the NIP individually and the set of these immunizers. Identifying the factors associated with the completeness of the vaccination schedule may assist services and governments in enhancing their efforts to achieve the desired level of vaccination coverage.

## 2. Materials and Methods

### 2.1. Study Design and Location

This was a single-center cross-sectional study of all cases of PLHIV recorded in the Notifiable Diseases Information System (NODINS) for HIV/AIDS among residents of Ribeirão Preto, São Paulo, Brazil. At the time of the study, its public health network included 50 basic units/health centers (36 of them with vaccine rooms) and 5 specialized care services (SCS). As for the SCSs, one of them had a vaccine room, three operated in the same physical area as basic units with vaccine rooms, and one did not have a vaccine room in the same physical space [18].

### 2.2. Population and Sample

The study population was composed of all cases recorded by the NODINS for HIV/AIDS infection in people aged 13 or over, between 2015 and 2020, in the municipality of Ribeirão Preto/SP, who met the following inclusion criteria: individuals who were being followed up with in the public health system, living in the municipality of Ribeirão Preto/SP, with a date of HIV diagnosis 180 days or less from the date of notification. In the NODINS, HIV/AIDS in adults is recorded for individuals over the age of 13. 

Individuals registered in the information system from 2015 onward were selected for inclusion in the study given that, at the time that the project was developed (2018), the most recent iteration of the vaccination schedule for PLHIV had been published in the latter half of 2014.

Therefore, patients registered in the system from this point forward were evaluated for their vaccination status following this schedule. It was decided that individuals diagnosed up to six months before the date chosen for HIV case records should be included to obtain a more homogeneous sample.

The exclusion criteria were as follows: cases that were recorded and followed up with by the supplementary health network; patients who were followed up with in the public network but not in the SCS, as they may have received vaccine doses through other services (private vaccination clinics or SIRCs), which would not have been registered in the municipality’s information system (Hygia-RP system); cases that were recorded in Ribeirão Preto but resided in another city; patients who died; and patients who were transferred to other services (supplementary network or other municipalities).

All patients registered in the information system during the chosen period were assessed regarding the inclusion and exclusion criteria and all those who met these criteria were included in the study. No sample calculation was carried out.

### 2.3. Data Collection

Sociodemographic variables were collected from the NODINS, which consolidates the registration of notifiable diseases and illnesses throughout the country. Clinical information on CD4 T-lymphocyte counts was collected from the Laboratory Examination Control System (LECSYS) and information on vaccine records was obtained from the Hygia-RP System and the National Immunization Program Information System (IS-NIP Web). These platforms (NODINS, LECSYS, and IS-NIP web) are systems linked to the UHS, whose purpose is to notify and account for the health information of each person/citizen through a single record. As such, they are systems used to collect epidemiological and statistical health data nationwide.

### 2.4. Vaccination Scheme

According to the Brazilian guidelines for PLHIV, vaccination coverage indicators refer to the percentage of individuals who have received adequate vaccination schedules for the indicated vaccines. Table 1 shows the vaccination schedule proposed by the NIP at the time of this study [19].

The ACWY meningococcal vaccine was incorporated into the PLHIV vaccination schedule in 2023 and the HPV vaccination schedule was extended to men up to 45 years of age in 2022, after the data collection of this study.

The following vaccines were not included in the evaluation of this study: varicella (only indicated for susceptible people, so it was not possible to assess susceptibility as this information was not included in the medical records), Haemophilus influenzae type b (the indication for this vaccine was not clear in the technical document and there was often a lack of immunobiologicals at the SIRC), and influenza (most immunized patients had no record of this vaccine’s administration in the system, since it is a vaccine used in seasonal campaigns).

### 2.5. Predictors of Vaccination

To assess the predictors of vaccination (outcome), independent variables of a sociodemographic, clinical–epidemiological, and social vulnerability nature were considered. 

Sociodemographic data: sex (male, female); age group (10 to 19, 20 to 29, 30 to 39, 40 and over); skin color (white, black, brown/yellow). 

Clinical–epidemiological data: specialized care service where PLHIV receive monitoring (SCS 1, 2, 3, 4, and 5), exposure to HIV category (heterosexual, non-heterosexual); social vulnerabilities (yes, no); TCD4+ lymphocyte count (<200, 200 to 350, >350 cells/mm^3^); ART adherence (yes, no); and the vaccination facility where the last vaccine doses were given (SCSs, other health services). 

Social vulnerability was defined as having a medical record of alcoholism, drug addiction, homelessness, and/or imprisonment.

### 2.6. Vaccination Coverage (Outcome)

The primary outcome was being adequately vaccinated for each vaccine indicated. The vaccination status of each subject was assessed for a range of vaccines, including those for diphtheria/tetanus, hepatitis B, hepatitis A, measles/mumps/rubella, yellow fever, 13- and 23-valent pneumococcal, meningococcal C, and HPV, following the schedule illustrated in Table 1.

Those who had received all recommended vaccines in accordance with the established vaccination schedule at the time of the evaluation, without any delays, were classified as having an “adequate schedule”. 

The models were adjusted by incorporating all independent variables, including gender, age group, skin color, social vulnerability, TCD4+ lymphocyte count, adherence to antiretroviral therapy, HIV exposure category, service responsible for follow-up, and the vaccination facility where the last vaccine doses were administered.

In this way, vaccination coverage was assessed dichotomously as “adequate” or “not adequate”, according to the national immunization schedule for PLHIV proposed by the NIP at the time of data collection. 

### 2.7. Data Analysis

Data were presented as frequencies and percentages, and associations were assessed using the chi-square or Fisher’s exact tests. Data on the adequacy of vaccination status for each of the nine vaccines evaluated individually and for all of them were examined using multivariate logistic regression, expressed as adjusted odds ratios (ORs) with a 95% confidence interval (95% CI), to compare the proportional differences in factors associated with vaccination status. The regression controlled for the effect of the independent variables adjusted for all other variables present in the model. A *p*-value < 0.05 was deemed to be statistically significant. 

Once the modeling had been completed, the Hosmer–Lemeshow goodness-of-fit test was applied to assess its suitability. This statistical procedure follows a chi-square distribution, aiming to verify the quality of the fit of a logistic regression model. It compares an event’s observed and expected frequencies across different risk groups. No discrepancy was observed between the anticipated and observed values (see Appendix A) [20,21]. 

The Wald test, used to ascertain the statistical significance of the respective parameters (betas) of the independent variables in a logistic regression model, was employed to evaluate the regression parameters. All analyses were conducted using the R programming language [22,23], version 4.3.1 (R software, developed by the R Core Team, Vienna, Austria).

### 2.8. Ethical Aspects

The project was approved by the Research Project Evaluation Committee (RPEC) of the Ribeirão Preto Municipal Health Department and by the Research Ethics Committee (REC) of the Ribeirão Preto College of Nursing of the University of São Paulo (USP), under number 4.782.341. 

## 3. Results

Figure 1 illustrates the composition of the study sample, while Table 2 presents a descriptive and multivariate regression analysis of the adequacy of the vaccines administered to the participants.

A total of 645 PLHIV were included in the study (Figure 1). Most of them were male (538; 83.4%) and white (386; 59.8%), with a minimum age of 14 and a maximum of 72. The most common age group was 20 to 29 years (296; 45.89%), followed by 30 to 39 years (173; 26.8%) (Table 2). 

Of the total, only 47 individuals (7.29%) had an adequate vaccination schedule for the nine vaccines according to the assessment proposed in this study. The adequate vaccination schedule was less prevalent among women (1.87%; *p* = 0.014), among those with a CD4 count < 200 cells/mm^3^ (1.74%; *p* = 0.015), and among those who did not have good adherence to ART (0.96%; *p* = 0.003) (Table 2). 

As for the age group, the best vaccination rate was identified among the youngest individuals aged between 10 and 19 (12.82%; *p* = 0.005), and no individual aged 50 or over had an adequate vaccination schedule. The adequacy of the vaccination schedule was associated with having received the last doses of vaccines at the SCS (*p* = 0.009) and with having been identified in the homosexual exposure to HIV category (*p* < 0.001). There was no difference in the status of the vaccination schedule concerning the health unit where the follow-up was carried out, social vulnerability, or skin color (Table 2).

In the multivariate analysis, individuals aged 40 or over had an 87.00% lower chance (aOR: 0.13, 95%CI: 0.02–0.72; *p* = 0.020) of being adequately vaccinated compared to individuals aged 10 to 19. Concerning the health unit where they received their last doses of the vaccine, individuals vaccinated at the SCS were 10 times more likely to be adequately vaccinated when compared to individuals who received their last doses of the vaccine at other health facilities (aOR: 10.44, 95%CI: 1.35–80.83; *p* = 0.025) (Table 2).

Figure 2 illustrates the percentage of individuals who had received adequate vaccination for each vaccine included in the study. Among inactivated vaccines that do not contain live organisms and are not contraindicated in cases of immunosuppression, the one with the highest rate of adequate vaccination was the diphtheria and tetanus vaccine (533; 82.6%), while the hepatitis A vaccine had the lowest number (144; 41.7%). Among the attenuated vaccines that contain live organisms and are contraindicated in cases of severe immunosuppression, most individuals (529; 82.0%) were adequately vaccinated against yellow fever and only a few of them (243; 37.7%) against measles/mumps/rubella (Figure 2) [13].

All patients (n = 645) were considered eligible to receive the double adult, 13-valent pneumococcal, 23-valent pneumococcal, meningococcal C, measles/mumps/rubella, and yellow fever vaccines. Patients with a last TCD4+ lymphocyte count < 200 cells/mm^3^ were not excluded from this indication. The number of individuals who met the criteria for other vaccines with a special recommendation (excluding those who were not susceptible or who were outside the age range for indication) was 594 for hepatitis B, 345 for hepatitis A, and 248 for HPV (Figure 2).

In the multivariate analysis, individuals aged between 20 and 29 exhibited a 62% lower chance of receiving inactivated vaccines (aOR: 0.38, 95% CI: 0.16–0.95; *p* = 0.038), and those aged between 40 and 49 exhibited an 82% lower chance of adequate hepatitis A vaccination (aOR: 0.18, 95% CI: 0.05–0.63; *p* = 0.007) than those aged between 10 and 19 (Figure 3).

Those aged between 30 and 39 had an 82% lower chance (aOR: 0.18, 95%CI: 0.04–0.85; *p* = 0.031) of being adequately vaccinated with the HPV vaccine (Figure 4) and individuals aged 40 or over were twice as likely (aOR: 2.47, 95%CI: 1.1–5.55; *p* = 0.028) to be adequately vaccinated with the 23-valent pneumococcal vaccine (Figure 5) than those aged between 10 and 19.

In comparison to white individuals, those who had brown or yellow skin exhibited a 58% reduced probability of being adequately vaccinated against hepatitis A (aOR: 0.42, 95% CI: 0.23–0.77; *p* = 0.005) (Figure 3) and a 40% reduced probability of being adequately vaccinated with the 23-valent pneumococcal vaccine (aOR: 0.6, 95% CI: 0.41–0.9; *p* = 0.013) (Figure 5). 

Those who were not socially vulnerable exhibited a twofold increased chance of being adequately vaccinated against hepatitis A (aOR: 2.79, 95% CI: 1.46–5.33; *p* = 0.002) (Figure 3) and a 1.9 times increased likelihood of being adequately vaccinated with the meningococcal C vaccine than individuals who exhibited social vulnerability (aOR: 1.91, 95% CI: 1.2–3.04; *p* = 0.007) (Figure 6). 

Individuals with TCD4+ lymphocyte counts < 200 cells/mm^3^ had a 44% lower likelihood of having received the 23-valent pneumococcal vaccine (aOR: 0.56, 95%CI: 0.35–0.89; *p* = 0.014) (Figure 5) and a 43% lower likelihood of having been adequately vaccinated against meningococcus C than those with TCD4+ lymphocyte counts > 350 cells/mm^3^ (aOR: 0.57, 95% CI: 0.34–0.96; *p* = 0.035) (Figure 6).

Those who demonstrated sustained adherence to ART exhibited a twofold increased chance of being adequately vaccinated against diphtheria and tetanus (aOR: 2.19, 95%CI: 1.19–4.01; *p* = 0.011) (Figure 7) and receiving the 13-valent pneumococcal vaccine (aOR: 2.59; 95%CI: 1.56–4.31; *p* = 0.001) (Figure 8) than those with poor ART adherence.

With regard to the SCSs, SCS 2 had a vaccine room; SCS 1, 3, and 4 operated in the same physical area as basic units with vaccine rooms; and SCS 5 did not have a vaccine room in the same physical space.

Individuals who were followed up in SCS 3 were three times more likely to be adequately vaccinated against diphtheria and tetanus (aOR: 3.32, 95%CI: 1.3–8.47; *p* = 0.012) (Figure 7) and twice as likely to be adequately vaccinated against hepatitis B (aOR: 2.16, 95%CI: 1.07–4.36; *p* = 0.031) (Figure 9). They also had a 71% lower chance (aOR: 0.29, 95%CI: 0.11–0.77; *p* = 0.012) of being adequately vaccinated with the HPV vaccine (Figure 4) and a 64% lower chance (aOR: 0.36, 95%CI: 0.2–0.65; *p* = 0.001) of being adequately vaccinated with the 13-valent pneumococcal vaccine (Figure 8) than those followed up with in SCS 2. 

Individuals who were followed up with at SAE 4 were twice as likely to be adequately vaccinated against hepatitis B (aOR: 2.09, 95%CI: 1.06–4.15; *p* = 0.034) (Figure 9) and five times more likely to be adequately vaccinated against hepatitis A (aOR: 5.64, 95%CI: 2.65–12; *p* = 0.001) (Figure 3) and had a 50% lower chance (aOR: 0.5, 95%CI: 0.28–0.9; *p* = 0.020) of being adequately vaccinated with the 13-valent pneumococcal vaccine (Figure 8) than those followed up with at SCS 2.

Individuals who were followed up with at SCS 1 had a 58% lower chance (aOR: 0.42, 95%CI: 0.24–0.74; *p* = 0.002) and those followed up with at SCS 5 had a 74% lower chance (aOR: 0.26, 95%CI: 0.14–0.46; *p* = 0.001) of being adequately vaccinated with the 13-valent pneumococcal vaccine (Figure 8) when compared to individuals who were followed up with at SCS 2.

Those who had received their last doses of vaccines at SCSs were three times more likely to be vaccinated against hepatitis B (aOR: 3.3, 95%CI: 1.92–5.67; *p* = 0.001) (Figure 9) and against hepatitis A (aOR: 3.48, 95%CI: 1.57–7.71; *p* = 0.002) (Figure 3) and were twice as likely to be adequately vaccinated with the 13-valent pneumococcal vaccine (aOR: 2.64, 95%CI: 1.61–4.33; *p* = 0.001) (Figure 8) and 23-valent pneumococcal vaccine (aOR: 2.15, 95%CI: 1.31–3.52; *p* = 0.002) (Figure 5) and four times more likely (aOR: 4.87, 95%CI: 2.9–8.19; *p* = 0.001) to be adequately vaccinated with the meningococcal C vaccine (Figure 6) when compared to those who received the last doses of the vaccines at other health units.

Among the attenuated vaccines, those aged between 30 and 39 had a 94% and those aged 40 or over had a 99% (aOR: 0.06, 95%CI: 0.02–0.14; *p* = 0.001 and aOR: 0.01, 95%CI: 0–0.04; *p* = 0.001) lower chance of being adequately vaccinated with the measles/mumps/rubella vaccine (Figure 10), and those aged between 30 and 39 were two times more likely (aOR: 2.96, 95%CI: 1.18–7.42; *p* = 0.021) to be adequately vaccinated with the yellow fever vaccine (Figure 11) than those aged between 10 and 19.

Black individuals were three times more likely (aOR: 3.44, 95%CI: 1.17–10.13; *p* = 0.025) to be adequately vaccinated with the yellow fever vaccine than white individuals (Figure 11).

Those with any T-lymphocyte CD4 count < 200 cells/mm^3^ had a 53% (aOR:0.47, 95%CI: 0.26–0.84; *p* = 0.011) lower chance of being adequately vaccinated with the yellow fever vaccine when compared to individuals with all counts > 350 cells/mm^3^ (Figure 11). 

## 4. Discussion

The present study sought to evaluate the predictors of the completeness of the vaccination schedule among PLHIV in Brazil. Overall, our findings indicate that a relatively small proportion of PLHIV in the municipality under study had an adequate vaccination schedule for the nine vaccines evaluated. The main predictors of the completeness of the vaccination schedule were the age group, the place of clinical follow-up, and where the last doses of the vaccines were received. Our evidence contributes to the understanding of the factors that should be considered when drawing up a vaccination care plan for PLHIV. It also provides evidence that can be used to design and strengthen government actions to achieve greater vaccine coverage. 

In Brazil, the recommendation is that 95% of individuals with an indication for vaccination receive the full course of the indicated vaccine [24]. In our study, no vaccine neared this target. Only the double adult (diphtheria/tetanus) and yellow fever vaccines had been administered in just over 80% of individuals with a complete or ongoing vaccination schedule without delays. These two vaccines have comprised the national vaccination schedule for the entire population for decades, without particularities in the recommendation schedule for PLHIV. Our participants might have completed the schedule for these two vaccines before they started being followed up with by HIV care facilities.

On the other hand, the vaccines with the lowest rates of adequate coverage were the triple viral and hepatitis A vaccines. The first one is available to the general population but has a different schedule for PLHIV, with a recommendation of two doses regardless of the age group. The other one is not included in the vaccination schedule for the general adult population in the country but has recently been included in the children’s recommended schedule. 

In the study by Neto, Vieira, and Ronchi (2017), also carried out in Brazil with PLHIV, the best coverage was also identified for the adult double vaccine, with 59.79% of individuals having a complete vaccination schedule. The hepatitis A vaccine was among the vaccines with the lowest coverage, with only 6.8% of individuals having a complete schedule. The authors indicated that the hepatitis A vaccine was not included in the vaccination schedule for PLHIV before 2014. Consequently, when the study started data collection in January 2015, the health teams had not incorporated this indication into their healthcare practice yet, and the immunizer supply was limited to Reference Centers for Special Immunobiologicals. This resulted in the potential under-coverage of this vaccine at the time of the evaluation [11].

In Ceará, Brazil, Cunha and colleagues (2016) showed that 83.8% of PLHIV had not received the triple viral vaccine, while only 7.07% had a record of yellow fever vaccination. Not being instructed about the vaccine and its regimens and not knowing the benefits of immunization were among the main reasons for non-adherence to vaccination [25]. 

The vaccination rate in PLHIV for the adult double vaccine in the Czech Republic was 97.4%, and the rate of those adequately vaccinated against hepatitis A was even lower than that identified in our study, at 29.0% [26].

In Ribeirão Preto, where this study took place, the hepatitis A and 13- and 23-valent pneumococcal vaccines are not openly available in vaccination rooms or to the general population. They must be requested via medical prescription and by filling in a specific form at the referral service. Another potential complicating factor is the absence of communication between the teams responsible for monitoring patients and the teams administering vaccinations. These latter teams may be hesitant to proceed with vaccinations in the absence of a doctor’s prescription. Additionally, the teams must utilize active search strategies to locate and vaccinate patients who have fallen behind in their schedules [6,11,27,28,29].

The adequacy of the vaccination status for the nine vaccines evaluated showed a positive association for those who received their last vaccines in services where PLHIV are followed up with, i.e., in specialized care services. The adequacy of vaccination for hepatitis B, hepatitis A, 13- and 23-valent pneumococcal, and meningococcal C was likewise positively correlated with the receipt of the last scheduled doses of the vaccines at the SCSs. These teams appear to be more highly trained and better informed about the recommended vaccination schedules for PLHIV, particularly vaccines unavailable to the general population. Training and equipping healthcare teams in primary healthcare vaccination rooms is crucial to ensure that they can also provide an updated vaccination status in accordance with the current recommendations, particularly for vaccines with special indications for PLHIV. 

For some vaccines, such as the adult double vaccine, hepatitis B, and hepatitis A, the facility where the clinical follow-up was carried out showed a positive association with the adequacy of the vaccination status. For the 13-valent pneumococcal and HPV vaccines, the place where clinical follow-up was carried out showed a negative association with the adequacy of the vaccination status.

Assessing patients’ vaccination status is not common practice in services that follow up with PLHIV, as the teams may have other priorities during their care [28,30]. Checking patients’ vaccination status and indicating the recommended vaccines should be part of the routines of such services since the medical prescription of immunizers can have a positive impact on patients’ decisions to be vaccinated, as well as directing the care of the teams in the vaccination rooms [11,27,31].

The lack of guidance from health teams regarding the indicated vaccines and their schedules and patients’ lack of knowledge regarding the benefits of vaccination are factors that contribute to non-vaccination as well [7,25,31].

Low vaccination coverage in PLHIV has already been demonstrated in other studies, but none of them have evaluated the adequacy of the vaccination schedule and its predictive factors taking into account nine immunizers, including attenuated vaccines, contraindicated in the presence of severe immunodepression, and none of them have shown such low coverage when evaluating a group of vaccines indicated for PLHIV [6,10,11,12,26,28,30,31].

In our study, having good ART adherence was positively associated with an adequate vaccination status for the adult double vaccine and the 13-valent pneumococcal vaccine. These findings are similar to those identified in another study in Brazil, in which patients with poor adherence or a poor response to treatment were also less likely to follow vaccination recommendations [11].

A TCD4+ count lower than 200 cells/mm^3^ was negatively associated with the adequacy of the vaccination status for 23-valent pneumococcal, meningococcal C, and yellow fever, as observed in other studies. It is recommended that vaccination takes place early for inactivated vaccines, which do not contain living organisms, regardless of the TCD4+ count, so that the opportunity for vaccination is not missed, taking into account the greater vulnerability of this group [6,25,26,28,30,31]. 

The possibility of events supposedly attributable to vaccination or immunization cannot be a factor in delaying or unnecessarily contraindicating vaccination, as the benefits of immunizing this group have already been demonstrated in several studies [27,30].

It should be noted that only attenuated vaccines, which contain living organisms in their composition, are contraindicated under severe immunodepression, and health professionals need to be properly trained to indicate and administer vaccines to PLHIV without hesitation. In addition, patients should be aware of the role of vaccination and the vaccines indicated for them so that they do not hesitate in being vaccinated [25,29,31,32,33,34].

The adequacy of the vaccination status for the nine vaccines evaluated in this study showed a negative association with the age group of 40 years and over. No individual over 50 years had an adequate overall vaccination status following the national recommendations. Being in a higher age group was a factor that negatively interfered with the adequacy of the vaccination status for the HPV, hepatitis A, and triple viral vaccines. However, this association was positive for the 23-valent pneumococcal and yellow fever vaccines.

The NIP’s adult vaccination schedule recommends two doses of the triple viral vaccine for the population up to 29 years of age, while, for those over 30, the recommendation is for a single dose [13]. This can be a confusing factor for teams in vaccination rooms, as they may forget that the recommendation for PLHIV is two doses regardless of the age group, which means that the population over 30 is less likely to have an adequate vaccination schedule for this vaccine. Furthermore, this is an attenuated vaccine, which requires a medical prescription for its administration to PLHIV, which can also make access to the vaccine difficult.

Another important aspect is that the 23-valent pneumococcal vaccine is recommended for individuals with comorbidities, and, for many years, it was a vaccine with a strong recommendation for the over 60s. This is a factor that may influence the indication of vaccination for older people with other comorbidities, increasing the vaccination rate among individuals in the older age group.

In a study carried out in Germany with PLHIV over the age of 50, the coverage of the double adult, hepatitis B, and pneumococcal vaccines was close to that identified in our study. The authors reinforced the importance of the actions of health professionals in improving the vaccination rates, as they need to be adequately informed about the safety, efficacy, and importance of vaccination for PLHIV [30].

Age was also a relevant factor regarding the vaccination rates in other studies, in which older age groups have shown the worst vaccination rates [6,35,36]. Immunization actions around the world and the search for desirable vaccination coverage rates focus mainly on children, while actions that seek to adjust the vaccination rates in adults, including those with special indications for vaccination, are rarely developed [7,8].

Our findings, when considered alongside those of other studies, indicate that a lack of awareness among healthcare professionals regarding the recommended vaccination schedule for individuals with HIV and the absence of an assessment of their vaccination status during clinical follow-up may contribute to the low vaccination rates. Despite the limited data on the efficacy and immunogenicity of most vaccines for PLHIV, there is a consensus that they represent a crucial tool in preventing complications from vaccine-preventable diseases. Consequently, it is essential to enhance the vaccination rates, ensuring that every visit to the health service presents an opportunity to assess and update patients’ vaccination statuses [11,15,16,17,27].

Furthermore, healthcare professionals must counsel individuals on the benefits of vaccination, act against misinformation, and ensure that vaccines are available and easily accessible at locations where individuals undergo their clinical follow-ups [11,27].

Our findings should be interpreted in light of some limitations. The data were collected through the UHS data management system and therefore may not represent the vaccination profiles of PLHIV being monitored in the private sector. In addition, the data were collected at a municipal level and may not include vaccinations carried out in other municipalities or before the implementation of the electronic data recording system. Therefore, some findings may be underestimated. However, municipal collaboration is required to update the historical records of each patient at the first visit to the vaccination room, and our sample exceeded the minimum size stipulated for the strong representation of this population.

Additionally, it should be noted that the study was conducted in only one Brazilian municipality, which may limit the generalizability of the findings to other contexts with different social, demographic, and epidemiological profiles. This underscores the need for additional research to investigate whether there are variations in the vaccination coverage of PLHIV in different settings. 

## 5. Conclusions

The study showed that vaccination coverage among adults with HIV is low, falling short of the country’s target immunization rates. Vaccination rates can be influenced by factors linked to patients and the establishment of their clinical follow-ups. Training the health teams involved in PLHIV follow-up, as well as the teams from the vaccination rooms of primary health care services, would help them to understand the schedules indicated for the immunobiologicals made available to this group, as well as the indication and administration of these immunobiologicals, avoiding missed vaccination opportunities. In addition, it is crucial that PLHIV are also aware of the benefits of vaccination and the vaccines indicated for them so that they do not hesitate to be vaccinated.

## Figures and Tables

**Figure 1 vaccines-12-01003-f001:**
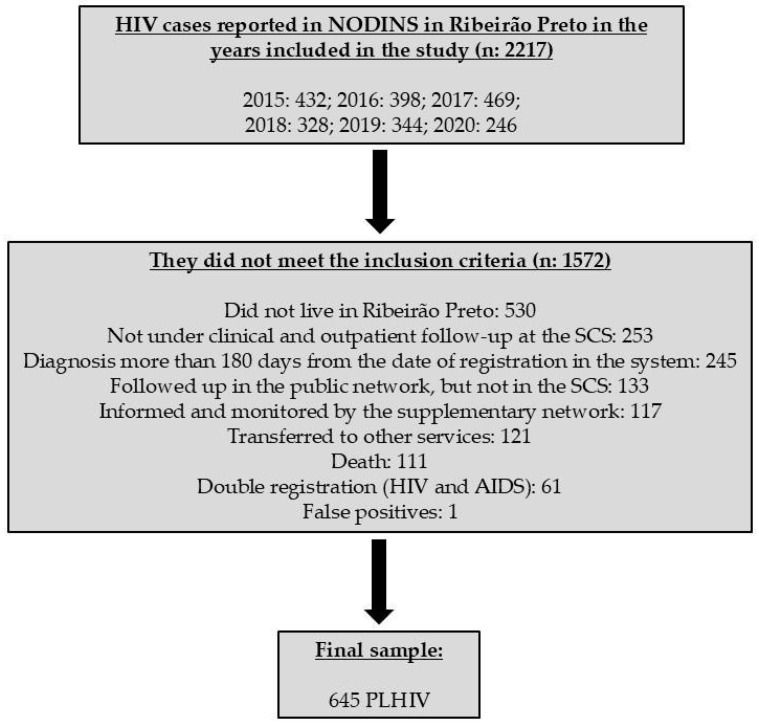
Flow diagram of the study sample construction. Ribeirão Preto, SP, Brazil, 2021.

**Figure 2 vaccines-12-01003-f002:**
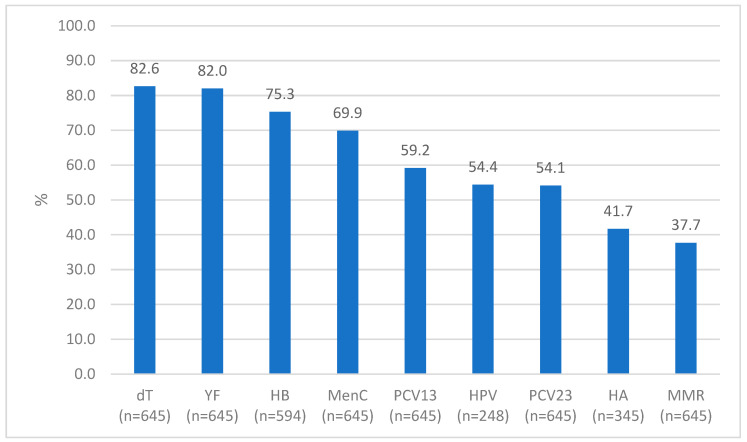
Percentage of PLHIV adequately vaccinated according to each vaccine. Ribeirão Preto, SP, Brazil, 2021. dT—diphtheria and tetanus; YF—yellow fever; HB—hepatitis B; MenC—meningococcal C; PCV13—13-valent pneumococcal; HPV—human papillomavirus; PCV23—23-valent pneumococcal; HA—hepatitis A; MMR—measles/mumps/rubella.

**Figure 3 vaccines-12-01003-f003:**
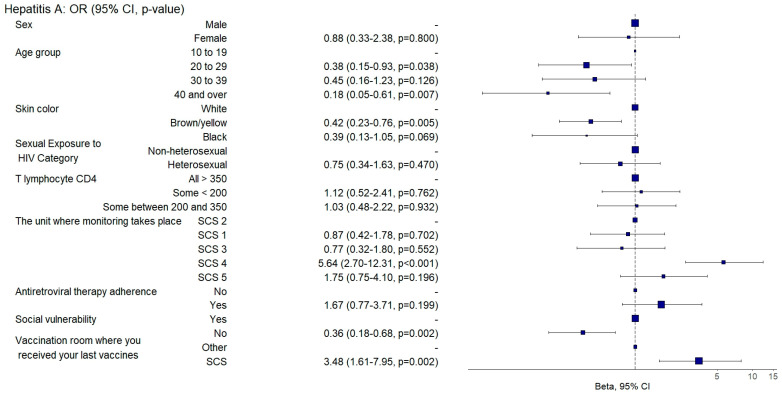
Analysis of hepatitis A vaccine adherence according to sociodemographic and clinical–epidemiological variables. Ribeirão Preto, SP, Brazil, 2021. SCS 2—specialized care service had a vaccine room; SCS 1, 3, 4—specialized care services operated in the same physical area as basic units with vaccine rooms; SCS 5—specialized care service did not have a vaccine room in the same physical space.

**Figure 4 vaccines-12-01003-f004:**
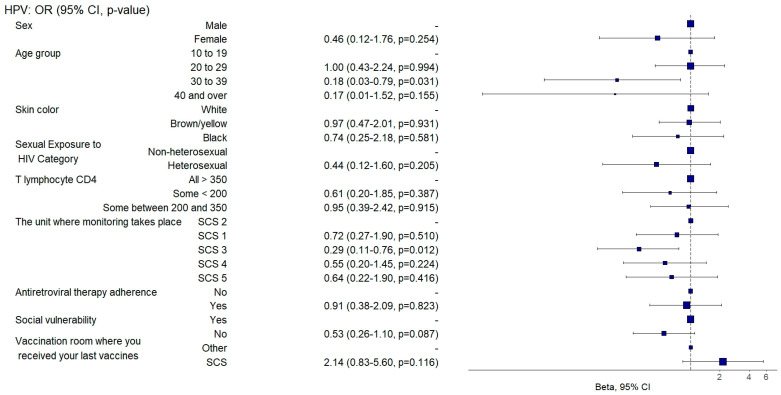
Analysis of HPV vaccine adherence according to sociodemographic and clinical–epidemiological variables. Ribeirão Preto, SP, Brazil, 2021. SCS 2—specialized care service had a vaccine room; SCS 1, 3, 4—specialized care services operated in the same physical area as basic units with vaccine rooms; SCS 5—specialized care service did not have a vaccine room in the same physical space.

**Figure 5 vaccines-12-01003-f005:**
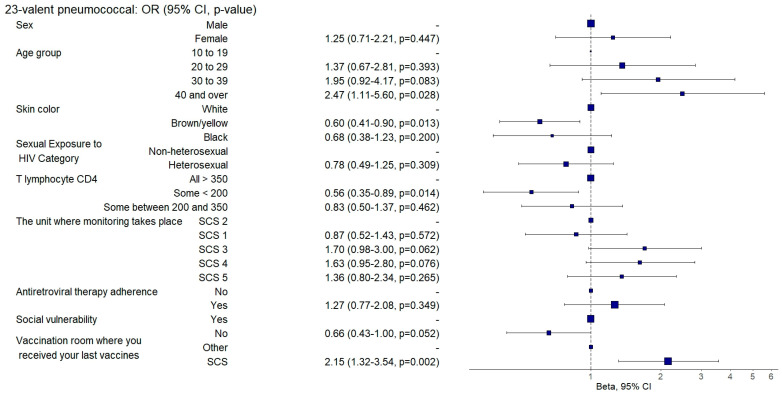
Analysis of 23-valent pneumococcal vaccine adherence according to sociodemographic and clinical–epidemiological variables. Ribeirão Preto, SP, Brazil, 2021. SCS 2—specialized care service had a vaccine room; SCS 1, 3, 4—specialized care services operated in the same physical area as basic units with vaccine rooms; SCS 5—specialized care service did not have a vaccine room in the same physical space.

**Figure 6 vaccines-12-01003-f006:**
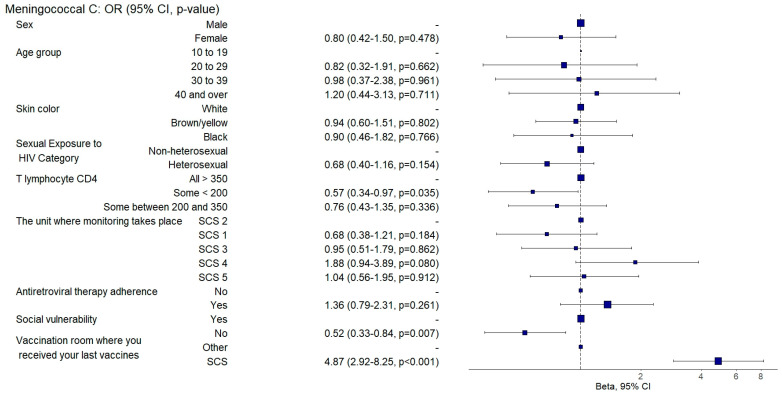
Analysis of meningococcal C vaccine adherence according to sociodemographic and clinical–epidemiological variables. Ribeirão Preto, SP, Brazil, 2021. SCS 2—specialized care service had a vaccine room; SCS 1, 3, 4—specialized care services operated in the same physical area as basic units with vaccine rooms; SCS 5—specialized care service did not have a vaccine room in the same physical space.

**Figure 7 vaccines-12-01003-f007:**
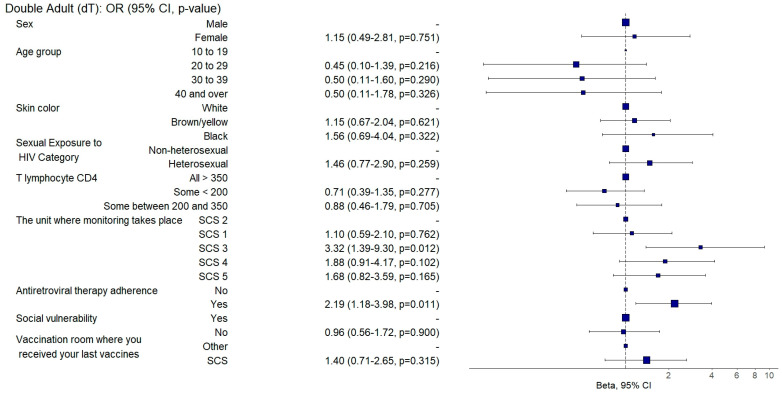
Analysis of double adult (dT) vaccine adherence according to sociodemographic and clinical–epidemiological variables. Ribeirão Preto, SP, Brazil, 2021. SCS 2—specialized care service had a vaccine room; SCS 1, 3, 4—specialized care services operated in the same physical area as basic units with vaccine rooms; SCS 5—specialized care service did not have a vaccine room in the same physical space.

**Figure 8 vaccines-12-01003-f008:**
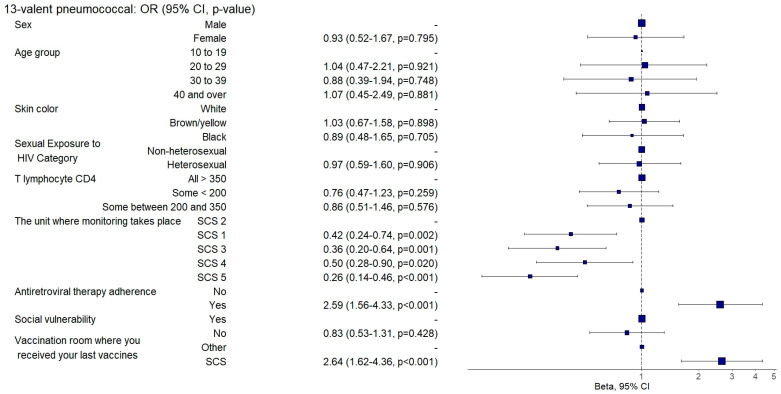
Analysis of 13-valent pneumococcal vaccine adherence according to sociodemographic and clinical–epidemiological variables. Ribeirão Preto, SP, Brazil, 2021. SCS 2—specialized care service had a vaccine room; SCS 1, 3, 4—specialized care services operated in the same physical area as basic units with vaccine rooms; SCS 5—specialized care service did not have a vaccine room in the same physical space.

**Figure 9 vaccines-12-01003-f009:**
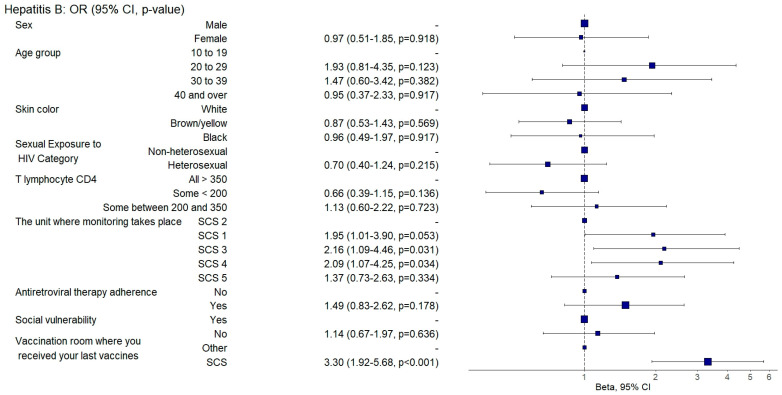
Analysis of hepatitis B vaccine adherence according to sociodemographic and clinical–epidemiological variables. Ribeirão Preto, SP, Brazil, 2021. SCS 2—specialized care service had a vaccine room; SCS 1, 3, 4—specialized care services operated in the same physical area as basic units with vaccine rooms; SCS 5—specialized care service did not have a vaccine room in the same physical space.

**Figure 10 vaccines-12-01003-f010:**
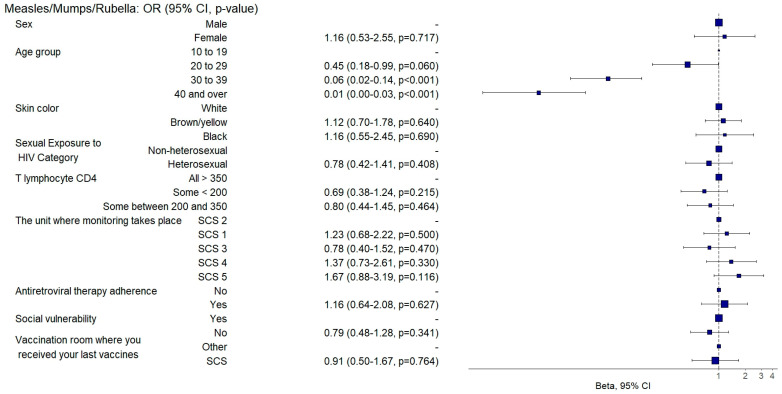
Analysis of measles/mumps/rubella vaccine adherence according to sociodemographic and clinical–epidemiological variables. Ribeirão Preto, SP, Brazil, 2021. SCS 2—specialized care service had a vaccine room; SCS 1, 3, 4—specialized care services operated in the same physical area as basic units with vaccine rooms; SCS 5—specialized care service did not have a vaccine room in the same physical space.

**Figure 11 vaccines-12-01003-f011:**
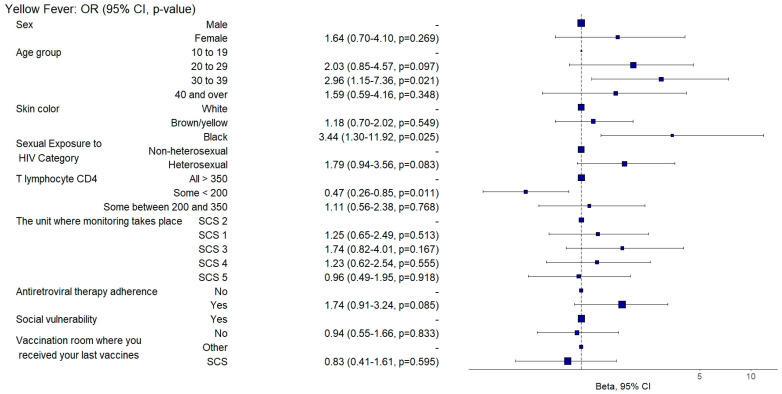
Analysis of yellow fever vaccine adherence according to sociodemographic and clinical–epidemiological variables. Ribeirão Preto, SP, Brazil, 2021. SCS 2—specialized care service had a vaccine room; SCS 1, 3, 4—specialized care services operated in the same physical area as basic units with vaccine rooms; SCS 5—specialized care service did not have a vaccine room in the same physical space.

**Table 1 vaccines-12-01003-t001:** Vaccines included in the assessment of adequate vaccination schedules according to the National Immunization Program (NIP) recommendations at the time of the study.

Vaccine	Recommended Schedule	Considered Schedule
Adult double (diphtheria/tetanus)	Three doses and boosters every 10 years	Complete or in progress without delay
Hepatitis B	Three doses (before diagnosis) Four doses with double dose (after diagnosis) ^†^ for susceptibles	Complete or in progress without delay
Hepatitis A	Two doses ^‡^ for susceptibles	Complete or in progress without delay
Measles, mumps, rubella (MMR)	Two doses	Complete or in progress without delay
Yellow fever	Single dose from 5 years of age or two doses	Complete or in progress without delay
23-valent pneumococcal	Two doses	1 dose
13-valent pneumococcal	Single dose	1 dose
Meningococcal C	Two doses and booster every 5 years	Complete or in progress without delay
HPV	Three doses (men aged 9 to 26 and women aged 9 to 45) ^§^	Complete or in progress without delay

^†^ Individuals with reactive anti-HBsAg serology without vaccine doses and those with reactive HBsAg or anti-HBc serology were excluded from the analysis for the hepatitis B vaccine. ^‡^ Individuals with reactive anti-HAV IgG serology were excluded from the analysis for the hepatitis A vaccine. ^§^ Data collection began on 1 August 2021, and the evaluation considered the vaccination schedule indicated by the NIP until this date, without incorporating subsequent updates.

**Table 2 vaccines-12-01003-t002:** Univariate and multivariate analysis for the appropriate vaccination schedule. Ribeirão Preto, SP, Brazil, 2021.

Sociodemographic and Clinical Variables	Adequate Scheme				
	Yes 47 (7.3%)	No 598 (92.7%)	*p*-Value	Adjusted OR (95% CI)	*p*-Value
Sex (n = 645)					
Male	45 (8.36%)	493 (91.64%)	**0.014** ^†^	*REF* ^‡^	
Female	2 (1.87%)	105 (98.13%)		0.26 (0.03–2.26)	0.223
Age group (n = 645)					
10 to 19	5 (12.82%)	34 (87.18%)	**0.005** ^†^	*REF* ^‡^	
20 to 29	32 (10.81%)	264 (89.19%)		0.58 (0.21–1.66)	0.312
30 to 39	8 (4.62%)	165 (95.38%)		0.32 (0.09–1.07)	0.064
40 to 49	2 (2.70%)	72 (97.30%)		0.13 (0.02–0.72) ^§^	0.020
50 to 59	0 (0.00%)	45 (100.00%)			
60 or more	0 (0.00%)	18 (100.00%)			
Skin color ^‖^ (n = 623)					
White	31 (8.03%)	355 (92.97%)	0.173 ^†^	*REF* ^‡^	
Black	2 (3.08%)	63 (96.92%)		0.50 (0.11–2.31)	0.376
Yellow	1 (33.33%)	2 (66.67%)		0.93 (0.45–1.90) ^§^	0.833
Brown	12 (7.10%)	157 (92.90%)			
Social vulnerability (n = 645)					
Yes	6 (3.85%)	150 (96.15%)	0.058 ^¶^	*REF* ^‡^	
No	41 (8.38%)	448 (91.62%)		1.67 (0.70–4.03)	0.250
T-lymphocyte CD4 count (cells/mm^3^) from diagnosis to current date ^‖^ (n = 644)					
All > 350	39 (9.00%)	394 (91.00%)	**0.015** ^†^	*REF* ^‡^	
Some < 200	2 (1.74%)	113 (98.26%)		0.29 (0.07–1.27)	0.100
Some between 200 and 350	6 (6.25%)	90 (93.75%)		0.82 (0.32–2.14)	0.691
Antiretroviral therapy adherence ^‖^ (n = 643)					
No	1 (0.96%)	103 (99.04%)	**0.003** ^†^	*REF* ^‡^	
Yes	46 (8.53%)	493 (91.47%)		6.00 (0.79–45.53)	0.083
Exposure to HIV category ^‖^ (n = 620)					
Homosexual	41 (11.85%)	305 (88.15%)	**<0.001** ^†^	*REF* ^‡§^	
Bisexual	0 (0.00%)	50 (100.00%)			
Other	0 (0.00%)	1 (100.00%)			
Heterosexual	6 (2.69%)	217 (97.31%)		0.52 (0.18–1.52)	0.234
Unit where follow-up is carried out (n = 645)					
Specialized Care Service 2 ^††^	17 (8.94%)	173 (91.05%)	0.078 ^†^	*REF* ^‡^	
Specialized Care Service 1 ^‡‡^	6 (4.84%)	118 (95.16%)		0.58 (0.21–1.57)	0.282
Specialized Care Service 3 ^‡‡^	2 (2.04%)	96 (97.96%)		0.42 (0.11–1.56)	0.194
Specialized Care Service 4 ^‡‡^	11 (10.09%)	98 (89.91%)		1.58 (0.66–3.80)	0.308
Specialized Care Service 5 ^§§^	11 (8.87%)	113 (91.13%)		1.81 (0.74–4.39)	0.192
Vaccination facility where last dose was received ^‖^ (n = 631)					
Basic health unit	1 (0.93%)	106 (99.07%)		*REF* ^‡§^	
Other	0 (0.00%)	2 (100.00%)	**0.009** ^†^		
Specialized care services	46 (8.81%)	476 (91.19%)		10.44 (1.35–80.83)	0.025

^†^ Fisher’s exact test; ^‡^ reference group; ^§^ in the regression, the variables were combined (40 to 49/50 to 59/ 60 or more; yellow/brown; homosexual/bisexual/other; basic health unit/other); ^‖^ data “not identified”/“exam not available”/“no information”/“not applicable”/“unknown” were not taken into account when processing the statistical test; ^¶^ Pearson’s chi-square test; ^††^ specialized care service had a vaccine room; ^‡‡^ specialized care services operated in the same physical area as basic units with vaccine rooms; ^§§^ specialized care service did not have a vaccine room in the same physical space; bolded values are significant at the 0.05 level. Source: Study data, Ribeirão Preto/SP, 2021.

## Data Availability

Data from this study can be made available upon request.

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
