# Peer review of "Factors Associated with Vaccination Adequacy in People Living with HIV: A Cross-Sectional Study"

_vaccines, 2024, doi:10.3390/vaccines12091003_

Round 1

Reviewer 1 Report (New Reviewer)

Comments and Suggestions for Authors

The article aims to identify predictors associated with adherence to recommended vaccination schedules among people living with HIV (PLHIV) in a large Brazilian municipality. While the study addresses an important public health issue, there are several critical aspects that need to be addressed for the findings to be considered robust and meaningful.

·       The study is described as a "single-center cross-sectional study," which inherently limits the generalizability of its findings. Conducting the study in a single municipality in Brazil restricts the demographic and socio-economic diversity of the sample, thus potentially skewing the results. Furthermore, the sample size of 645 individuals may not be adequate to represent the entire population of PLHIV, especially considering the diverse epidemiological profiles that exist in different regions.

·       The study mentions the use of multiple logistic regression to identify predictors of adequate vaccination schedules. However, it does not specify whether potential confounders were controlled for. The choice of independent variables is mentioned (sociodemographic, clinical-epidemiological, and social vulnerability factors), but there is no clarity on how these were measured or categorized.

·       The primary outcome, "adequacy of the vaccination schedule," is not clearly defined. The criteria used to determine adequacy for each vaccine and collectively across vaccines are not provided in the abstract. Moreover, the use of self-reported vaccination status could lead to recall bias, potentially affecting the accuracy of the findings.

·       The results are presented with odds ratios (OR) and 95% confidence intervals (CI), does not it specify the level of statistical significance. Without this information, it's challenging to assess the strength and reliability of the associations found.

·       While the study concludes that educational interventions are needed, it provides only a generic recommendation. There is no discussion of specific barriers to vaccination or tailored strategies that could address the unique needs of PLHIV.

·       Population Selection: May not capture variations in vaccination coverage among those living with HIV for longer periods.

·       Exclusion of Individuals Lost to Follow-up: May introduce selection bias, as those lost to follow-up could differ systematically in ways that affect vaccination uptake (e.g., access to healthcare, socioeconomic status).

·       Limited generalizability to other regions or populations, especially in areas with different healthcare infrastructure, socio-economic conditions, or epidemiological profiles.

·       Cross-sectional design limits the interpretation of causality; the results are associative and not indicative of causal relationship

Comments on the Quality of English Language

 English very difficult to understand/incomprehensible

Author Response

Dear reviewer,

               We would like to thank you for the suggestions made to the manuscript “Factors associated with vaccination adequacy in people living with HIV: a cross-sectional study”.

The author’s responses are listed below.

Coments 1: The study is described as a "single-center cross-sectional study," which inherently limits the generalizability of its findings. Conducting the study in a single municipality in Brazil restricts the demographic and socio-economic diversity of the sample, thus potentially skewing the results. Furthermore, the sample size of 645 individuals may not be adequate to represent the entire population of PLHIV, especially considering the diverse epidemiological profiles that exist in different regions.

Response 1: Thank you for bringing this to our attention. We have indeed included this limitation in our manuscript.

Coments 2: The study mentions the use of multiple logistic regression to identify predictors of adequate vaccination schedules. However, it does not specify whether potential confounders were controlled for. The choice of independent variables is mentioned (sociodemographic, clinical-epidemiological, and social vulnerability factors), but there is no clarity on how these were measured or categorized.

Response 2 - Right, we have explained it on subitem 2.5 in the manuscript.

Coments 3: The primary outcome, "adequacy of the vaccination schedule," is not clearly defined. The criteria used to determine adequacy for each vaccine and collectively across vaccines are not provided in the abstract. Moreover, the use of self-reported vaccination status could lead to recall bias, potentially affecting the accuracy of the findings.

Response 3: In the revised manuscript, we sought to provide greater clarity regarding the definition of the vaccination schedule considered in the study, as presented in Table 1. For further information, kindly requer to sub-item 2.6. It was not possible to present all the criteria used in the study in the abstract due to the maximum number of words allowed by the journal.

Also, we apologise if the manuscript was unclear at this point. Actually, the self-reported vaccination status was not employed as a data collection method in this study. Instead, the official municipal surveillance system was utilised for data gathering, as it records all administered vaccine doses.

Coments 4: The results are presented with odds ratios (OR) and 95% confidence intervals (CI), does not it specify the level of statistical significance. Without this information, it's challenging to assess the strength and reliability of the associations found.

Response 4 – A p-value of less than 0.05 was deemed to be statistically significant. This information can be found in sub-item 2.7. To facilitate comprehension, the information has been positioned in closer proximity to the OR and CI.

Coments 5: While the study concludes that educational interventions are needed, it provides only a generic recommendation. There is no discussion of specific barriers to vaccination or tailored strategies that could address the unique needs of PLHIV.

Response 5 – We have included the specific barriers to vaccination and strategies that can help improve PLHIV vaccination rates in the discussion.

Coments 6: Population Selection: May not capture variations in vaccination coverage among those living with HIV for longer periods.

Response 6: This study is part of a doctoral thesis in which a cohort study was conducted to ascertain the vaccination coverage of PLHIV at two distinct points in time, before and following an educational initiative targeting health professionals. To achieve feasibility, it was essential to ascertain the population's homogeneity. Consequently, the study included only individuals registered in the information system from 2015 onward, given that the national vaccination schedule for PLHIV was updated in the latter half of 2014, and the project was developed in 2018. The results demonstrate an increase in vaccination coverage after the educational activity. However, the data presented in this manuscript comprises the pre-intervention phase.

Also, we highlight that this is a relevant and unprecedented study in Brazil. The majority of studies identified regarding vaccination coverage of people living with HIV involve populations from developed and high-income countries. Brazil is a developing country with specific characteristics that researchers must take into account, including a public and universal health system that provides a wide range of free immunisers to the population. Nevertheless, the identified vaccination coverage rates were relatively low, emphasising the necessity for further studies on this subject.

Coments 7: Exclusion of Individuals Lost to Follow-up: May introduce selection bias, as those lost to follow-up could differ systematically in ways that affect vaccination uptake (e.g., access to healthcare, socioeconomic status).

Response 7: Thank you for your question. We do indeed understand the issue you have raised. We hope that the reasons for the exclusion criteria, which we outlined in our previous response, are now clear.

Coments 8: Limited generalizability to other regions or populations, especially in areas with different healthcare infrastructure, socio-economic conditions, or epidemiological profiles.

Response 8: Dear reviewer, I believe this comment has already been answered in Response 1.

Coments 9: Cross-sectional design limits the interpretation of causality; the results are associative and not indicative of causal relationship

Response 9: Dear reviewer, we appreciate your input and agree with your perspective. Throughout the text, we have intended to indicate an association between the variables. If there is any passage in which our writing/translation may have gone beyond the boundaries of our inference, we would be grateful if you could bring it to our attention.

Once again, we are grateful for your attention and availability during the manuscript review process and for all your valuable recommendations.

We look forward to receiving instructions on the next steps and remain at your disposal for any further explanation.

Best regards,

The authors.

Reviewer 2 Report (New Reviewer)

Comments and Suggestions for Authors

This paper reports a descriptive statistical analysis of cross-sectional sample data on covariates associated with vaccination adequacy in people living with HIV in a large Brazilian municipality. Overall, the study follows standard protocols for statistical analyses in such studies and the paper is relatively well written. Here are a couple of things to attend to in a revision.

First, in the paragraph that begins on line 186 of the text, it would be good to include a more detailed description of the Hosmer-Lemeshow and Wald tests and explain what each does and how they differ.

Second, on first citation, it would be good to introduce each of the tables and figures with a sentence like "Table 1 includes descriptive and multivariate regression statistics for adequacy of the vaccinations of sample members" follow by paragraphs verbally describing the details of the respective descriptive and multivariate entries in the tables. 

Author Response

Dear reviewer,

               We would like to thank you for the suggestions made to the manuscript “Factors associated with vaccination adequacy in people living with HIV: a cross-sectional study”.

The author’s responses are listed below.

Coments 1: in the paragraph that begins on line 186 of the text, it would be good to include a more detailed description of the Hosmer-Lemeshow and Wald tests and explain what each does and how they differ.

Response 1: Sure! We have added this information on subitem 2.7.

Coments 2: on first citation, it would be good to introduce each of the tables and figures with a sentence like "Table 1 includes descriptive and multivariate regression statistics for adequacy of the vaccinations of sample members" follow by paragraphs verbally describing the details of the respective descriptive and multivariate entries in the tables. 

Response 2: Thanks for your contribution. We’ve included this information according to your suggestion.

Once again, we are grateful for your attention and availability during the manuscript review process and for all your valuable recommendations.

We look forward to receiving instructions on the next steps and remain at your disposal for any further explanation.

Best regards,

The authors.

Reviewer 3 Report (New Reviewer)

Comments and Suggestions for Authors

Estimated Authors,

I've been invited to review your interesting study on the Factors Associated with Vaccination Adequacy in People Living with HIV (PLwHIV): a cross-sectional study. This is a very important topic, for several reasons. On the one hand, PLwHIV are at (obviously) high risk for developing severe complications after infections from considered pathogens. On the other hand, the availability of effective long-term therapy for HIV has raised the opportunity for guaranteeing appropriate vaccination strategies ALSO for immunodeficient people. According to this report, the current status of vaccinations among PLwHIV is far from optimal, with an overwhelming majority of subjects requiring to update their status. Regarding the factors associated with being or not being vaccinated, age groups and ethnicity were highly expected. Very interesting that factors associated with characteristics of SCS could radically impact on the vaccination rates.

Despite the substantial interest from this study, I will recommend some improvements, and more precisely:

1) please detail the design of logistic regression analysis: were all variable included as "a priori" explanatory variable of the outcome variable of positive vaccination status?  I guess so, but please state in materials and methods.

2) please change the term "skin color" to "ethnicity" and please refer to the international terminology

3) Have you considered to shift Tables 2 to 4 to a Forrest Plot? It would allow to directly compare the estimates from all assessed vaccinations.

4) Figure 2 should be moved in the early stages of the results, immediately after the description of the demography and immediately before the univariate analysis.

5) Figure 1 is of very low quality (in terms of resolution); please redesign for being more print-friendly.

Author Response

Dear reviewer,

               We would like to thank you for the suggestions made to the manuscript “Factors associated with vaccination adequacy in people living with HIV: a cross-sectional study”.

The author’s responses are listed below.

Coments 1: please detail the design of logistic regression analysis: were all variable included as "a priori" explanatory variable of the outcome variable of positive vaccination status?  I guess so, but please state in materials and methods.

Response 1: Sure. We gave more details about it on item 2.6.

Coments 2: please change the term "skin color" to "ethnicity" and please refer to the international terminology

Response 2: Dear reviewer, we recognise that this type of classification may be somewhat confusing outside Latin America. We are sorry for any inconvenience this may cause. The historical process of colonisation in Brazil has resulted in a population with a diverse range of ethnic backgrounds. Given this complexity, it is not straightforward to assess the ethnicity of participants in a specific way. Therefore, the Brazilian Institute of Geography and the Ministry of Health recommend assessing skin colour rather than ethnicity. Our data was collected from the country's health information systems; therefore, we followed the same classification.

Coments 3: Have you considered to shift Tables 2 to 4 to a Forrest Plot? It would allow to directly compare the estimates from all assessed vaccinations.

Response 3: We haven´t, but that was a good plan! We shifted the findings for a Forest Plot, and it's much better now! Thanks a lot for this suggestion!   

Coments 4: Figure 2 should be moved in the early stages of the results, immediately after the description of the demography and immediately before the univariate analysis.

Response 4: the figure 2 was organised as suggested.

Coments 5: Figure 1 is of very low quality (in terms of resolution); please redesign for being more print-friendly.

Response 5: We apologise for that. We updated the design, but let us know if there's anything else we can do to make it even better.

Once again, we are grateful for your attention and availability during the manuscript review process and for all your valuable recommendations.

We look forward to receiving instructions on the next steps and remain at your disposal for any further explanation.

Best regards,

The authors.

Reviewer 4 Report (New Reviewer)

Comments and Suggestions for Authors

Dear authors,

I have now completed the review of the manuscript titled "Factors Associated with Vaccination Adequacy in People Living with HIV: a cross-sectional study."

The manuscript is interesting and, in general, fairly well-written.

I have some suggestions to further improve the quality of the manuscript.

I would like to suggest that the authors address these limitations in the article, either by discussing them in the limitations section or, where feasible, by making the appropriate revisions:

1. In introduction and discussion, more previous research with similar topics should be discussed or compared. Since many Comprehensive Systematic Reviews were done on Immunogenicity of COVID-19 Vaccines in Patients with Diverse Health Conditions already.

2. The study was conducted in a single municipality in Brazil, so the findings may not be generalizable to other regions or countries with different healthcare systems and vaccination practices for people living with HIV (PLHIV). I would like to hear authors' idea to solve this.

3. The study used a very narrow definition of social vulnerability (alcoholism, drug addiction, homelessness, imprisonment). I wonder authors have further assessment of social vulnerability. A more comprehensive assessment of socioeconomic factors could reveal additional insights. 

Thank you for your valuable contributions to our field of research. I look forward to receiving the revised manuscript.

Author Response

Dear reviewer,

               We would like to thank you for the suggestions made to the manuscript “Factors associated with vaccination adequacy in people living with HIV: a cross-sectional study”.

The author’s responses are listed below.

Coments 1: In introduction and discussion, more previous research with similar topics should be discussed or compared. Since many Comprehensive Systematic Reviews were done on Immunogenicity of COVID-19 Vaccines in Patients with Diverse Health Conditions already.

Response 1: We agree with the proposal to introduce and discuss other research findings in this context.

Coments 2: The study was conducted in a single municipality in Brazil, so the findings may not be generalizable to other regions or countries with different healthcare systems and vaccination practices for people living with HIV (PLHIV). I would like to hear authors' idea to solve this.

 Response 2: Thank you for bringing this to our attention. We have indeed included this limitation in our manuscript, as you rightly observed.

Coments 3: The study used a very narrow definition of social vulnerability (alcoholism, drug addiction, homelessness, imprisonment). I wonder authors have further assessment of social vulnerability. A more comprehensive assessment of socioeconomic factors could reveal additional insights. 

 Response 3: Thank you for your valuable suggestion. Regrettably, this study did not conduct more in-depth assessments of social vulnerability, given the limitations of the data available in the national surveillance system. However, we concur that a more comprehensive assessment of socioeconomic factors could yield further insights.

Once again, we are grateful for your attention and availability during the manuscript review process and for all your valuable recommendations.

We look forward to receiving instructions on the next steps and remain at your disposal for any further explanation.

Best regards,

The authors.

Round 2

Reviewer 4 Report (New Reviewer)

Comments and Suggestions for Authors

All comments were addressed.

This manuscript is a resubmission of an earlier submission. The following is a list of the peer review reports and author responses from that submission.

Round 1

Reviewer 1 Report

Comments and Suggestions for Authors

Larissa Gerin et.al examined what factors influence whether people living with HIV (PLHIV) in Brazil complete their recommended vaccinations. The study found that very few people (only 7.3%) finished all their vaccines when looking at the data from over 600 participants. The study's findings that age, clinic location, and final vaccination site are important factors in completing the vaccination schedule are unsurprising. The research also highlights how important it is to improve vaccination for people living with HIV (PLHIV).

Strengths of the Study:

  1. The research clearly explains how they conducted the study (single-center retrospective cohort study) and where it took place (located 312 km from the capital in Brazil).
  2. The research carefully described the people involved, including who could and couldn't participate.
  3. They clearly explained how they collected data, using reliable sources.
  4. The types of vaccines and the detail information such as recommend/considered schedules included in the study were well-explained.
  5. They used appropriate statistical methods to analyze the data.

Suggestions for Improvement:

  1. Consider removing the sentence about distance and population size compared to the capital (Line 94). This doesn't directly affect the main results.
  2. Briefly mention how many participants they expected after applying the criteria, giving context to the final number. They should also explain why they chose 180 days as the cut-off for time since HIV diagnosis.
  3. Briefly state how the study was ethically approved. Including this information strengthens the research.

Author Response

Coments 1: Consider removing the sentence about distance and population size compared to the capital (Line 94). This doesn't directly affect the main results.

Response 1: Removed line 94.

Coments 2: Briefly mention how many participants they expected after applying the criteria, giving context to the final number. They should also explain why they chose 180 days as the cut-off for time since HIV diagnosis.

Response 2: Data regarding the number of participants were presented through a flowchart. The requested explanation regarding the choice of 180 days was made.

Coments 3: Briefly state how the study was ethically approved. Including this information strengthens the research.

Response 3: Completed the requested description.

Reviewer 2 Report

Comments and Suggestions for Authors

Title Factors associated with completing the vaccination schedule for people living with HIV

Thank you for the opportunity to review this manuscript.  The purpose of this study was to identify the predictors associated with completing the vaccination schedule for people living with HIV.  Overall, I think it is an important topic.  I hope the authors find my suggestions helpful.

Line 91-92 Is “study of all cases of PLHIV not notified” correct?  I don’t understand what this means. Also, is “notified” interchangeable with “documented”? 

Exclusion criteria – people followed up by the supplementary health network and patients who were followed up in the public network, but not in the SCSs,….  I think a flowchart would be helpful here (lines 101-113) or at the beginning of the results sections to show how many people who met the initial inclusion criteria dropped out due to the exclusion criteria.  I am having a hard time following who is in the sample and how representative they may be of the overall population in this municipality with HIV.

Chart 1. What does “for susceptibles” mean?  How is susceptibility defined for these vaccines?

Section 2.5

-          Ages in years is indicated here, but not used in any tables

-          Exposure category – exposure to HIV?  Should be explicit

-          Follow-up service (SCS 1-5)  Are these the locations where people with HIV follow up for their healthcare after being diagnosed with HIV?  Just HIV care, or overall healthcare?  You mentioned in the Introduction that 1 had a vaccine room, 3 were part of basic units with vaccine rooms, and one did not have a vaccine room.  It seems as though this could be highlighted in the analyses rather than just labeling 1-5.  Readers do not know which type the reference group (SCS=2) indicates.

-          What about year of diagnosis?  Inclusion criteria spanned 2015-2020.  Was each person followed for the same amount of time or were all people assessed at the time of the data pull, so those diagnosed in 2015 had several more years to become fully vaccinated than those in 2020? Also, would you expect vaccination rates to be different for those diagnosed in 2020 given the COVID-19 epidemic and as a result of the vaccination discussions/activities surrounding it?

Section 2.6

-          When was vaccination status assessed in relation to HIV diagnosis?  Does it matter? See my questions above.

Section 2.7

-          Lines 169-179 “In this analysis….non-heterosexual exposure category.” I don’t think this level of detail about comparison groups is necessary here, but rather the reader should be able to discern this from the results tables.

-          Were there missing values for the outcomes or independent variables?  If so, how was that handled?  Were people just dropped from the analyses if any missing values?

-          Please indicate whether or not you adjusted for multiple comparisons, and provide rationale for your decision.

-          Did you consider whether you had ample sample size (particularly for some of the vaccines with smaller sample sizes) to carry out logistic regressions with the number of independent variables you used?

Section 3. Results

-          As mentioned earlier, if a flowchart wasn’t included in inclusion/exclusion area, it would be helpful here to understand better who is represented in this sample.

-          Line 183 “645 PLHIV took part in the study” – awkward wording since subjects were presumably not consented for the research as this was just secondary data analysis.  I think it would be clearer to say “were included” rather than “took part”.

-          If you want to report average age, which I don’t think is necessary, you should indicate in the text that it is not reported in Table 1.

-          Table 1 (some comments are relevant to all tables)–

o   last column should be updated to English;

o   use of *, **, *** should be avoided if possible because they are generally used to indicate levels of significance in quantitative tables and here they are used differently;

o   all abbreviations should be defined in a note under the table;

o   I usually see “(REF)” as in “reference group” as opposed to “comparison category”;

o   if bolded values are significant at .05 level, you should indicate that in a note below the table;

o   significant values in the crude OR column are not bolded.  That said, I do not think the Crude analyses columns are necessary given the frequency columns significance tests to the left of them. 

o   The independent variable labels should all be left-justified.  Some are centered in the table.

o   OR and CIs should use consistent decimal places. 

o   The table should be interpretable without the text – by looking at the table alone, the reader does not know that some categories were collapsed for the logistic regression.  I had to refer back to the earlier text before the results section.

o   Please check the reference groups for all of your dichotomous variables sex, social vulnerability, ART. I’m not sure they are in the right directions.  Results look backwards.

-          Figure 1 –

o   order the vaccines by rate of adequate vaccination;

o   calibrate y axis to 100%;

o   label the y-axis;

o   define the abbreviations in a note below the table;

o   add n’s underneath each vaccine in horizontal axis

-          Beginning in line 217, the authors start to categorize the vaccinations by inactivated vs attenuated, but those terms were not defined earlier nor discussed as important.

-          Tables 2-4 are very busy.  I don’t think the crude OR columns are necessary.  Removing them will also make it easier for the Cis to fit next to the ORs.  In these tables, some confidence intervals underneath the OR, but others don’t and that make it more difficult to read. Please refer to my general comments about Table 1 as well. 

-          What are the n’s for each vaccine analysis?  Could be indicated in table.

-          Rather than focusing on each individual vaccine when laying out the results of tables 2-4, it might be more helpful to discuss results in terms of the independent variable effects across the vaccines.  Are there patterns? 

-          As with table 1, please check the coding of the ART, sex, and social vulnerability variables.  Neither the results for social vulnerability nor ART adherence make intuitive sense.  Given the percentages in the first columns of table 1, your results may be backwards.

Discussion

-          Please consider your discussion in light of the potential error in your ART, sex, and social vulnerability variables.

-          The flow of the discussion is awkward in spots.  E.g., 4 consecutive paragraphs begin with “In another study…”, “In another study”,  “In a study”, and “In the municipality of the study”.

Other minor comments

-          There are several places beginning in Author Contributions section through the References where “sea” appears to be replaced by “SCS”.

-          Line 73 – “PLHIV is” – should be “PLHIV are”

-          Line 74 – missing “vaccines” after “immunobiological”?

Comments on the Quality of English Language

Overall, English was very good.  There were a few words/phrases that I think are not common that made it just a little challenging.

Author Response

Coments 1: Line 91-92 Is “study of all cases of PLHIV not notified” correct?  I don’t understand what this means. Also, is “notified” interchangeable with “documented”?

Response 1: All cases were reported to the surveillance system. We have corrected the term in the text to make it clearer. Yes, “notified” is interchangeable with “documented”.

Coments 2: Exclusion criteria – people followed up by the supplementary health network and patients who were followed up in the public network, but not in the SCSs,….  I think a flowchart would be helpful here (lines 101-113) or at the beginning of the results sections to show how many people who met the initial inclusion criteria dropped out due to the exclusion criteria.  I am having a hard time following who is in the sample and how representative they may be of the overall population in this municipality with HIV.

Response 2: The flowchart has been included at the beginning of the results section, as suggested by the reviewer. It is not possible to know the general population of this community living with HIV, as HIV infection only became mandatory information for health systems in the country in 2014, so many cases diagnosed before then were not registered.

Coments 3: Chart 1. What does “for susceptibles” mean?  How is susceptibility defined for these vaccines?

Response 3: Below the table is a description of susceptible individuals.

Coments 4:

Section 2.5

-          Ages in years is indicated here, but not used in any tables

-          Exposure category – exposure to HIV?  Should be explicit

-          Follow-up service (SCS 1-5)  Are these the locations where people with HIV follow up for their healthcare after being diagnosed with HIV?  Just HIV care, or overall healthcare?  You mentioned in the Introduction that 1 had a vaccine room, 3 were part of basic units with vaccine rooms, and one did not have a vaccine room.  It seems as though this could be highlighted in the analyses rather than just labeling 1-5.  Readers do not know which type the reference group (SCS=2) indicates.

-          What about year of diagnosis?  Inclusion criteria spanned 2015-2020.  Was each person followed for the same amount of time or were all people assessed at the time of the data pull, so those diagnosed in 2015 had several more years to become fully vaccinated than those in 2020? Also, would you expect vaccination rates to be different for those diagnosed in 2020 given the COVID-19 epidemic and as a result of the vaccination discussions/activities surrounding it?

Response 4:

- Deleted age in years from the section 2.5;

- We have corrected the exposure category.

- SAE 1-5 are specialised health services that provide follow-up care for PLHIV after they have received an HIV diagnosis. These units also provide follow-up care for other infectious diseases, including the prevention and treatment of tuberculosis, syphilis, chlamydia, gonorrhoea, and vaccinations. Further details on this topic can be found in the results section, where it is described in more detail.

- The data collection took place in 2021 and all patients were assessed in the same period. People who were enrolled in the system in 2015 had more time to complete their vaccination schedule; however, those who were in the process of completing their vaccination schedule were also considered to have an adequate vaccination schedule if they had no delays in their vaccination schedule. Therefore, we did not only consider those who had completed the vaccination schedule at the time of data collection. During the Covid-19 pandemic, vaccination routines were not interrupted in the municipality studied.

Coments 5:

Section 2.6

-          When was vaccination status assessed in relation to HIV diagnosis?  Does it matter? See my questions above.

Response 5: Vaccination status was not assessed by date of diagnosis or registration in the information system. In Brazil, the latest version of the vaccination calendar for PLHIV was published at the end of 2014, so patients registered in the system from 2015 would have their vaccination status assessed based on this calendar. It was decided to include people diagnosed up to 6 months before the date chosen for the HIV case records in order to have a more homogeneous sample. This information has been included in section 2.2.

Coments 6:

Section 2.7

-          Lines 169-179 “In this analysis….non-heterosexual exposure category.” I don’t think this level of detail about comparison groups is necessary here, but rather the reader should be able to discern this from the results tables.

-          Were there missing values for the outcomes or independent variables?  If so, how was that handled?  Were people just dropped from the analyses if any missing values?

-          Please indicate whether or not you adjusted for multiple comparisons, and provide rationale for your decision.

-         Did you consider whether you had ample sample size (particularly for some of the vaccines with smaller sample sizes) to carry out logistic regressions with the number of independent variables you used?

Response 6:

- Lines 169-179 were deleted.

- We excluded the missing data (marked as 'ignored') for the following variables to perform the regression analysis: CD4 T lymphocyte count (cells/mm3) from diagnosis to current date (n = 1), adherence to antiretroviral therapy (n = 1), vaccination room where the patient received the last vaccine dose (n = 14), skin colour (n = 22), type of HIV exposure (n = 25). Individuals who had no indication for some vaccines were excluded from the analysis during the vaccination schedule evaluation: hepatitis B (n = 52), hepatitis A (n = 298), HPV (n = 398).  

- Regression is naturally an analysis procedure in which the interpretation of a model parameter already considers the presence of other variables, as well as the adjusted OR.

- Table 1 answers this question showing up the total study population by applying the inclusion/exclusion criteria. No sample calculation was performed due to the recruitment procedure adopted.

Coments 7:

Section 3. Results

-          As mentioned earlier, if a flowchart wasn’t included in inclusion/exclusion area, it would be helpful here to understand better who is represented in this sample.

-          Line 183 “645 PLHIV took part in the study” – awkward wording since subjects were presumably not consented for the research as this was just secondary data analysis.  I think it would be clearer to say “were included” rather than “took part”.

-          If you want to report average age, which I don’t think is necessary, you should indicate in the text that it is not reported in Table 1.

Response 7:

- The flowchart was included in the results section as suggested.

- The suggested change was made.

- The average age was removed from the text.

Coments 8:

Table 1 (some comments are relevant to all tables):

-   last column should be updated to English;

-   use of *, **, *** should be avoided if possible because they are generally used to indicate levels of significance in quantitative tables and here they are used differently;

-   all abbreviations should be defined in a note under the table;

-   I usually see “(REF)” as in “reference group” as opposed to “comparison category”;

-   if bolded values are significant at .05 level, you should indicate that in a note below the table;

-   significant values in the crude OR column are not bolded.  That said, I do not think the Crude analyses columns are necessary given the frequency columns significance tests to the left of them. 

-   The independent variable labels should all be left-justified.  Some are centered in the table.

-   OR and CIs should use consistent decimal places. 

-   The table should be interpretable without the text – by looking at the table alone, the reader does not know that some categories were collapsed for the logistic regression.  I had to refer back to the earlier text before the results section.

- Please check the reference groups for all of your dichotomous variables sex, social vulnerability, ART. I’m not sure they are in the right directions.  Results look backwards.

Response 8:

Table 1:

- the correction was made;

- the symbols used were replaced;

- all abbreviations are inserted in a note below the tables;

- the term “comparison category” was replaced;

- indicated the information below the table as requested;

- excluded the columns from the raw analysis as suggested;

- correction made;

- In the paper, the p-values have 3 decimals, which is sufficient given the 5% significance level (alpha=0.05), and the OR and CI need no more than 2 decimals to be interpreted.

- we inserted a note below the table informing the combination of variables in the logistic regression;

- foi feita correção na apresentação das variáveis ​​dicotômicas, elas foram alteradas conforme observação do revisor.

Comentários 9 :

Figura 1:

- ordenar as vacinas por taxa de vacinação adequada;

- calibrar o eixo y para 100%;

- rotular o eixo y;

- definir as abreviaturas em nota abaixo da tabela;

- adicione n abaixo de cada vacina no eixo horizontal

Resposta 9:

Todas as sugestões relativas à Figura 1 foram atendidas.

Comentários 10 :

- Começando na linha 217, os autores começam a categorizar as vacinações em inativadas vs. atenuadas, mas esses termos não foram definidos anteriormente nem discutidos como importantes.

Resposta 10:

Os termos foram definidos na seção Resultados.

Comentários 11:

- As tabelas 2-4 são muito ocupadas. Não acho que as colunas OR brutas sejam necessárias. Removê-las também facilitará o ajuste do Cis ao lado dos ORs. Nessas tabelas, alguns intervalos de confiança abaixo do OR, mas outros não, o que dificulta a leitura. Consulte também meus comentários gerais sobre a Tabela 1. 

- Quais são os n's para cada análise de vacina? Pode ser indicado na tabela.

- Em vez de se concentrar em cada vacina individual ao apresentar os resultados das tabelas 2 a 4, poderá ser mais útil discutir os resultados em termos dos efeitos das variáveis ​​independentes entre as vacinas. Existem padrões? 

- Assim como na tabela 1, verificar a codificação das variáveis ​​TARV, sexo e vulnerabilidade social. Nem os resultados relativos à vulnerabilidade social nem à adesão à TARV fazem sentido intuitivamente. Dadas as percentagens nas primeiras colunas da tabela 1, os seus resultados podem estar invertidos.

Resposta 11:

As correções solicitadas foram feitas.

Comentários 12:

Discussão

- Por favor, considere sua discussão à luz do possível erro em suas variáveis ​​de TAR, sexo e vulnerabilidade social.

- O fluxo da discussão é estranho em alguns pontos. Por exemplo, 4 parágrafos consecutivos começam com “Em outro estudo…”, “Em outro estudo”, “Em um estudo” e “No município do estudo”.

Resposta 12:

As correções solicitadas foram feitas.

Comentários 13:

Outros comentários menores

- Existem vários lugares começando na seção Contribuições do Autor através das Referências onde “mar” parece ser substituído por “SCS”.

- Linha 73 – “PLHIV é” – deveria ser “PLHIV são”

- Linha 74 – faltando “vacinas” depois de “imunobiológicos”?

Resposta 13:

- Fizemos as correções sugeridas.

Round 2

Reviewer 2 Report

Comments and Suggestions for Authors

I appreciate the authors' attempts to address my initial concerns about the manuscript.  I think, however, that the manuscript still requires improvement. 

For example,

 - the n's reported in Figure 2 are the numerator n's of the proportions when they should be the denominators. 

- I apologize, but I am still confused by the first sentences in 2.1 and 2.2 – I've bolded the confusing language, which seems contradictory in the two sentences. 

 This is a single-center retrospective cohort study of all cases of PLHIV not notified in the Notifiable Diseases Information System (NODINS) for HIV/AIDS among residents of Ribeirão Preto, São Paulo, Brazil.

 and 

The study population was made up of all the cases informed to NODINS for  HIV/AIDS infection in people aged 13 or over between 2015 and 2020 in the municipality of Ribeirão Preto/SP  

The n's indicated in table 1 show that missingness on any given variable ranges from 0% - 4%, and so this could mean that more than 4% of the sample had at least 1 missing value.  The note says that these were excluded in the analyses, but the multivariate tables still show instances of n=645, which cannot be accurate.  What are the actual n's in each of the models?  For the main model, did the authors check to see whether this changed the overall composition of the sample that is reported?  

- In addition, the tables could be better organized - e.g., with sociodemographic variables grouped together, person clinical variables grouped, etc.  

- the results are hard to follow, and the discussion does not tie the results together in a very constructive way.  Overall, I think the paper needs better framing.  In my initial review, I had suggested perhaps reporting results by characteristic.  In the revision the authors did try to do this, but reported on tables 2 and 3 separately.  Unless there was a reason that particular vaccines were separated out into two tables (other than the number of vaccines) , the author could talk about tables 2 and 3 together.  Alternatively, in the introduction, there is mention of different vaccines that are recommended for the population in general, and those that are further recommended for PLHIV.  Later in the manuscript, the authors distinguish between attenuated and inactivated vaccines.  Neither of these two groupings (general/HIV  or attenuated/inactivated) are used to frame the presentation of the results or discussion in a meaningful way, but perhaps could be.  

 - when presenting the SCS in the table and text, it is hard to follow  when they are just labeled 1-5.  It would be more informative to talk about them in relation to the type of SCS.  This is especially true since the discussion seems to emphasize type of SCS in some places.

On a smaller note,

 - the decimal places are still inconsistent in the tables - I am not talking about using 2 for ORs and 3 for CIs - that is fine.  I mean that when reporting ORs, e.g., be consistent and report 2 decimal places for each item.  See table 2, e.g., for several places where only 1 decimal place is shown.  1.15 (0.4-2.72); 1.1 (0.59-2.07);  etc.

- crude ORs are still referenced in 2.7 but the authors removed them from the tables.

Comments on the Quality of English Language

Overall, the quality of English is good, though some phrasing/word choice differs.

Author Response

Dear reviewer,

               We would like to thank you for the suggestions made to the manuscript “Factors Associated with completing the vaccination schedule for people living with HIV”.

Below are responses to the reviewer's comments.

Coments 1: the n's reported in Figure 2 are the numerator n's of the proportions when they should be the denominators.

Response 1: the correction was made.

Coments 2: I apologize, but I am still confused by the first sentences in 2.1 and 2.2 – I've bolded the confusing language, which seems contradictory in the two sentences. 

 This is a single-center retrospective cohort study of all cases of PLHIV not notified in the Notifiable Diseases Information System (NODINS) for HIV/AIDS among residents of Ribeirão Preto, São Paulo, Brazil.

 and 

The study population was made up of all the cases informed to NODINS for  HIV/AIDS infection in people aged 13 or over between 2015 and 2020 in the municipality of Ribeirão Preto/SP  

Response 2: the correction was made in the first sentence in 2.1.

Coments 3: The n's indicated in table 1 show that missingness on any given variable ranges from 0% - 4%, and so this could mean that more than 4% of the sample had at least 1 missing value.  The note says that these were excluded in the analyses, but the multivariate tables still show instances of n=645, which cannot be accurate.  What are the actual n's in each of the models?  For the main model, did the authors check to see whether this changed the overall composition of the sample that is reported?  

Response 3: the actual n’s was reported in each variable. The n remained relevant and did not affect the fit the models. Missing data was treated as missing in their respective variables.

Coments 4: In addition, the tables could be better organized - e.g., with sociodemographic variables grouped together, person clinical variables grouped, etc.  

Response 4: the tables were organized as suggested.

Coments 5: the results are hard to follow, and the discussion does not tie the results together in a very constructive way.  Overall, I think the paper needs better framing.  In my initial review, I had suggested perhaps reporting results by characteristic.  In the revision the authors did try to do this, but reported on tables 2 and 3 separately.  Unless there was a reason that particular vaccines were separated out into two tables (other than the number of vaccines) , the author could talk about tables 2 and 3 together.  Alternatively, in the introduction, there is mention of different vaccines that are recommended for the population in general, and those that are further recommended for PLHIV.  Later in the manuscript, the authors distinguish between attenuated and inactivated vaccines.  Neither of these two groupings (general/HIV or attenuated/inactivated) are used to frame the presentation of the results or discussion in a meaningful way, but perhaps could be.  

Response 5: The results were reported according to the characteristics, describing tables 2, 3 and 4 separated by inactivated and attenuated vaccines. We really appreciate your comment and included attenuated vaccines in the introduction, results and discussion, which in our opinion improved the quality of the manuscript. In the discussion, vaccines specifically indicated for PLHIV were highlighted, with emphasis on attenuated vaccines that may be erroneously contraindicated for this population.

Coments 6: when presenting the SCS in the table and text, it is hard to follow  when they are just labeled 1-5.  It would be more informative to talk about them in relation to the type of SCS.  This is especially true since the discussion seems to emphasize type of SCS in some places.

Response 6: Information about SCSs was inserted in the table and text.

Coments 7:  - the decimal places are still inconsistent in the tables - I am not talking about using 2 for ORs and 3 for CIs - that is fine.  I mean that when reporting ORs, e.g., be consistent and report 2 decimal places for each item.  See table 2, e.g., for several places where only 1 decimal place is shown.  1.15 (0.4-2.72); 1.1 (0.59-2.07);  etc.

Response 7: the correction was made.

Coments 8:  crude ORs are still referenced in 2.7 but the authors removed them from the tables.

Response 8: the crude OR was removed in 2.7.

Once again, we are grateful for your attention and availability during the manuscript review process and for all your valuable recommendations.

We look forward to receiving instructions on the next steps and remain at your disposal for any further explanation.

Best regards,

The authors.
